# Cinnamaldehyde Enhances Antimelanoma Activity through Covalently Binding ENO1 and Exhibits a Promoting Effect with Dacarbazine

**DOI:** 10.3390/cancers12020311

**Published:** 2020-01-29

**Authors:** Weiyi Zhang, Jie Gao, Chuanjing Cheng, Man Zhang, Wenjuan Liu, Xiaoyao Ma, Wei Lei, Erwei Hao, Xiaotao Hou, Yuanyuan Hou, Gang Bai

**Affiliations:** 1State Key Laboratory of Medicinal Chemical Biology, College of Pharmacy and Tianjin Key Laboratory of Molecular Drug Research, Nankai University, Tianjin 300353, China; zwy13132150812@163.com (W.Z.); gaojie@nankai.edu.cn (J.G.); 2120171157@mail.nankai.edu.cn (C.C.); 18202572363@163.com (M.Z.); liuwenjuan00001@163.com (W.L.); mxy83396376@163.com (X.M.); l_wei8910@163.com (W.L.); 2Collaborative Innovation Center of Research on Functional Ingredients from Agricultural Residues, Guangxi Key Laboratory of Efficacy Study on Chinese Materia Medica, Guangxi University of Chinese medicine, Nanning 530200, China; ewhao@163.com (E.H.); houxt@126.com (X.H.)

**Keywords:** cinnamaldehyde, α-enolase (ENO1), covalent inhibitor, dacarbazine, antimelanoma

## Abstract

At present, melanoma is a common malignant tumor with the highest mortality rate of all types of skin cancer. Although the first option for treating melanoma is with chemicals, the effects are unsatisfactory and include poor medication response and high resistance. Therefore, developing new medicines or a novel combination approach would be a significant breakthrough. Here, we present cinnamaldehyde (CA) as a potential candidate, which exerted an antitumor effect in melanoma cell lines. Chemical biology methods of target fishing, molecular imaging, and live cell tracing by an alkynyl–CA probe revealed that the α-enolase (ENO1) protein was the target of CA. The covalent binding of CA with ENO1 changed the stability of the ENO1 protein and affected the glycolytic activity. Furthermore, our results demonstrated that dacarbazine (DTIC) showed a high promoting effect with CA for antimelanoma both in vivo and in vitro. The combination improved the DTIC cell cycle arrest in the S phase and markedly impacted melanoma growth. As a covalent inhibitor of ENO1, CA combined with DTIC may be beneficial in patients with drug resistance in antimelanoma therapy.

## 1. Introduction

Melanoma is a highly malignant tumor originating from melanocytes, and its incidence has been rising over the past 20 years, accounting for >80% of skin-cancer-related deaths [1]. This disease is characterized by an inconspicuous onset, high malignancy, metastasis, and a poor prognosis [2]. In addition to traditional surgical treatment, radiotherapy and chemotherapy are also used. Unfortunately, many medicines for melanoma have limitations [3]. Temozolomide (TMZ) is an alkylating agent that can be used to treat metastatic melanoma. Moreover, the widely studied TMZ resistance mechanisms are mainly related to the DNA repair capacity of the carcinoma cells [4]. Fotemustine (FTM), a nitrosourea analog, has shown modest activity in patients with metastatic melanoma, but its response rates were also approximately 20% [5]. Dacarbazine (DTIC), a monofunctional alkylating agent, has been approved as a chemotherapeutic medicine for melanoma treatment [6,7]. It exerts antitumor effects via the methylation of nucleic acids or direct DNA damage and leads to a G2/M phase arrest in melanoma cells [8,9]. However, it produces relative responses in only 15–20% of patients [10]. Studies have shown that cotreatment with DTIC and statins could suppress the growth of tumors and simultaneously improve the survival rate by suppressing the RhoA/RhoC pathway and inducing the expression of cleaved caspase-3, cleaved PARP, p21, p27, and p53 in melanoma [11]. Previous studies have also demonstrated that TMZ can enhance the antitumor activity of FTM, but they have limited effectiveness [12,13,14].

In recent years, targeted therapy and immunotherapy have obvious therapeutic effects on melanoma, but they are limited in the implementation process [15]. For example, pembrolizumab and nivolumab are PD1 inhibitors, they can enhance the rates of response and overall survival in melanoma patients compared with chemotherapy and ipilimumab (a CTLA4 inhibitor). Although these medicines have received much attention, 40–60% of patients still do not respond to them [16]. Therefore, there is a need to find new medicines or combination therapy approaches and additional effective therapeutic strategies for melanoma.

Cinnamaldehyde (CA) is an active ingredients that originates from cinnamon. It has been widely used as a natural flavorant and fragrance agent in the food processing, perfume, and pharmaceutical industries. It is assumed to be a safe natural ingredient agent and is well tolerated in human and animals [17,18]. Previous studies have reported that CA has many effects, including sedation, reducing hypertension, analgesia, antipyresis, anticonvulsion, antiobesity, anti-inflammatory, etc. [19,20,21,22]. In addition, CA can treat various types of cancer, including leukemia, hepatocellular carcinoma, and colon, breast, prostate, and oral cancers [23,24,25,26]. Only a few studies have reported that CA can inhibit the growth of melanoma cells and tumor growth [27], but the mechanism and target protein are not clearly explained.

It has been reported that chemical biological methods can reveal the antitumor effect of various natural products. For instance, Wang et al. identified nucleolin as curcumol’s novel molecular target in nasopharyngeal carcinoma [28]. Liu et al. demonstrated that adenanthin could induce the differentiation of acute promyelocytic leukemia (APL)-like cells, suppress tumor growth in vivo, and prolong the survival rate of mouse APL models by targeting peroxiredoxins I and II to induce cell differentiation [29]. Moreover, Lindsey et al. certified that TRPA1-activating compounds are electrophiles that can react with cysteines. For example, the nucleophilic mercapto group of cysteines can attack the α,β-unsaturated bond of CA by Michael addition. The covalent modification of reactive cysteines within TRPA1 can induce channel activation and rapidly signal potential tissue damage through the pain pathway [30]. In general, chemical biological methods can help us to better find the target and mechanism of action of medicines.

In this study, we designed an alkynyl-CA probe (AL-CA) to find the target of CA on melanoma, and colocalization imaging was applied to verify the key target. The results demonstrated that CA could arrest the cell cycle through α-enolase (ENO1) and eventually inhibit melanoma growth. Subsequently, the relationship between the combined administration of CA and a variety of antimelanoma medicines was explored. We found that CA could produce promoting effects with DTIC, and it significantly improved the therapeutic efficacy both in vitro and in vivo. These results may provide a new potential strategy for the treatment of melanoma.

## 2. Results

### 2.1. CA Inhibited A375 Cell Growth In Vitro and In Vivo

To verify the specificity of CA against different kinds of cells, multiple types of tumor cells and normal cells were tested. As shown in Figure 1A and Appendix A, compared with other cells, CA showed a better inhibitory effect in human malignant melanoma cell lines. Among them, A375 cells were chosen for the following experiments. Moreover, CA inhibited the proliferation of A375 cells in both a concentration- and a time-dependent manner (Figure 1B). The 50% growth inhibition concentration (IC_50_) value of CA against A375 cells was approximately 31.06 μM at 72 h. The proliferation inhibitory activity of CA was then further confirmed by colony-forming assay. As shown in Figure 1C, CA could significantly reduce the number of colonies in a dose-dependent manner. The trans-well migration assay showed that CA notably inhibited the cellular migration ability compared to that of the vector group in A375 cells (Figure 1D). To further evaluate the role of CA on tumor growth in vivo, a xenograft model was examined via A375 cells. As shown in Figure 1E, compared with the vector group, CA could also dose-dependently inhibit the growth of the tumors (detailed data provided in Appendix A).

To reveal the structure-activity relationship of CA, CA derivatives, cinnamic acid and 3-phenylpropanal were selected to evaluate their antimelanoma effects (Figure 1F). As shown in Figure 1G, CA showed marked cytotoxicity, but the two derivatives showed less toxicity in A375 cells. This phenomenon indicated that the α, β-unsaturated aldosterone in the structure of CA played a key role [31]. 

### 2.2. CA Covalently Bonded to ENO1 Protein

To identify the potential targets, an Al-CA probe was synthesized (Figure 2A). The IC_50_ of the Al-CA probe (IC_50_ = 53.25 μM) in A375 cells was slightly less than the value of CA (IC_50_ = 31.06 μM), and it could still induce apoptosis and cycle arrest at the G2/M/S phase (Appendix A). The result indicated that the Al-CA probe could be used in chemical biology testing of CA. Next, the Al-CA-functionalized magnetic microspheres (AL-CA-MMs) were prepared and used to capture the protein targets from the A375 cells (Figure 2A). The captured proteins were released by DTT reduction. After that, sodium dodecyl sulfate polyacrylamide gel electrophoresis (SDS-PAGE) was performed to detect the fishing efficiency (Figure 2B). As shown in Figure 2B (lane 3), one distinct band at about 50 kDa was detected. The captured protein was then recovered, enzymatically hydrolyzed, and identified by high perfromance liquid chromatography-tandem mass spectrometry (HPLC-MS/MS). The results showed that the highest score was from ENO1 protein. It had eight periods of sequence (red background) and was consistent with human ENO1 (Appendix A). Furthermore, Western blotting was used to confirm that the ENO1 target was significantly captured by AL-CA-MMs, compared with a non-Al-CA-modified microspheres group (Figure 2B).

CETSA indicated that the covalent binding changed the stability of the protein of ENO1 in a temperature- and dose-response manner (Figure 2C,D). The silenced ENO1 affected the glycolytic process of 2-phospho-D-glycerate (2PG) to phosphoenolpyruvate (PEP), which was detected using an ENO1 Human Activity Assay Kit (ab117994, Abcam, (Cambridge, MA, USA), (Figure 2E). Hence, ENO1 was suggested as a potential antimelanoma target of CA.

### 2.3. CA Colocalized with ENO1 in both A375 Cells and Living Tumors

It was found that the ENO1 genes in humans and mice were highly homologous (94%) (Appendix A). To verify whether ENO1 is the target of CA, a fluorescence tracing test was designed and performed in both human melanoma cells and living mouse tumor tissues (Figure 3A). After treatment of the A375 cells with the Al-CA probe, the total protein was extracted and electrophoretically separated on SDS-PAGE, and then the click reaction was performed with an N_3_-tag in situ. As shown in Figure 3B, only ENO1 protein was detected on the gel. To evaluate the interaction between the Al-CA probe and ENO1 protein at the cellular level, cytoimmunohistochemistry was then performed. As shown in Figure 3C, the vector group and CA (10 μM) group showed little fluorescence for the CA molecule tracing. However, specific fluorescence (pseudo green) was observed in the Al-CA probe (10 μM) treatment group. Additionally, the ENO1 protein that was stained with Anti-ENO1 antibodies (pseudo red) was partially colocalized with the Al-GA probe (yellow). In the localization test on the tumor-bearing model, the mice were orally administered CA (60 mg/kg) and Al-CA (60 mg/kg) for two weeks. Then, the melanoma was sequentially removed, fixed, sectioned, and reacted with the Anti-ENO1 antibodies or/and fluorescence N_3_-tag, respectively. Compared with the minimal fluorescence of CA group, the pseudo green fluorescence of the Al-CA probe was observed in the tumor tissues, which merged partially with ENO1 protein antibody staining (red) and presented a colocalized yellow fluorescence. Besides this, the fluorescence sensitivity statistical result showed a high overlapping (Appendix A). The above evidence suggested that CA targets ENO1 both in vitro and in vivo.

### 2.4. CA Induced Apoptosis and Arrested the G2/M/S Phase in A375 Cells

To investigate whether CA could arrest the cell cycle, the cell cycle distribution was evaluated by a flow cytometer. As shown in Figure 4A, CA markedly induced G2/M/S phase arrest in A375 cells in a concentration-dependent manner. To detect whether CA induced apoptosis in A375 cells, Annexin V-FITC/PI double staining was performed, and the results showed that apoptotic cells were increased from 3.3% to 23.4% (Figure 4B). Furthermore, we examined several apoptosis-related proteins. We found that after CA treatment, antiapoptotic proteins including Bcl-2 and Bid decreased significantly, while the proapoptotic protein Bax increased (Appendix A). These results certified that both apoptosis and G2/M/S phase arrest triggered by CA contributed to the cytotoxicity in A375 cells.

To investigate the effect of the *ENO1* gene on the function of A375 cells, we silenced ENO1 and then examined the sensitivity to CA. As shown in Figure 4C, ENO1 mRNA and protein expression in the siENO1 group were significantly decreased compared to the negative control (NC) group. Additionally, A375 cells showed less sensitivity to CA after ENO1 silencing. Furthermore, silenced ENO1 reversed the G2/M phase arrest and induced G0/G1 phase arrest in A375 cells (Figure 4D). These results indicated that CA could affect the cell cycle process by inhibiting the function of ENO1, thus inducing apoptosis in A375 cells.

### 2.5. CA Combined with DTIC Promoted Antimelanoma Effect In Vitro and In Vivo

To verify the promoting effect of CA and other antimelanoma medications, the A375 cells were incubated with different concentrations of CA alone or in combination with FTM, DTIC, paclitaxel (TAX), TMZ, or cisplatin (Pt). The tested medicine concentrations were close to their IC_50_ values (Appendix A). We found that DTIC and TAX significantly increased the inhibition rate in A375 cells compared to the untreated controls. Furthermore, DTIC and TAX showed synergistic effects with CA and enhanced the inhibitory effect of CA on A375 cells (Appendix A). However, TAX exhibited high toxicity to cells. In addition, FTM and Pt had little effect and showed antagonistic effects at most concentrations (Appendix A). 

Subsequently, to detect the optimum effect of DTIC combined with CA in different kinds of melanoma cells, different concentrations of CA were tested for combined administration with DTIC. The statistics indicated that the synergistic effect could be achieved in a certain concentration range in vitro. CA enhanced the IC_50_ values of DTIC in A375, A875, C918, and SK-MEL-1 cells (Figure 5A–C, Appendix A). Furthermore, changes of the cell cycle after the combination of CA with DTIC was detected. The results showed that DTIC could enhance CA arrest in S phase (Figure 5D and Appendix A).

To evaluate the role of CA combined with DTIC in the promotion of inhibition of melanoma growth in vivo, the effects in an A375 ectopic melanoma model were further examined. As shown in Figure 5E, CA or DTIC inhibited the growth of the A375 xenografts, but no significant change in body weight was observed. In addition, their combination exhibited an enhanced effect. These data suggest that DTIC plays a promoting role with CA by arresting the cell cycle, thus enhancing its antimelanoma effect in vivo and in vitro.

## 3. Discussion

Human melanoma cells usually express distinct differences in their degrees of cell morphology, pigmentation, and growth. Due to this phenotypic heterogeneity, melanoma patients generally appear medicine-resistant during treatment and show a terrible response to conventional radiation [32,33]. Therefore, the discovery of new therapeutic chemicals and treatment strategies is a great challenge. In this paper, we demonstrated that CA could covalently combine with ENO1 to play an antimelanoma role. Additionally, a better strategy was suggested for combination therapy with DTIC against melanoma, both in vitro and in vivo.

Enolases (ENOs) are glycolytic enzymes that are responsible for the ATP-generated conversion of 2PG to PEP. Studies have identified three isoforms of enolases, α-, β-, and γ-enolase (ENO1, 2, 3) [34,35]. ENO1 is a critical glycolytic enzyme that plays a functional role in some physiological processes [36]. Some studies have reported that ENO1 expression increases on cell surfaces in response to various stimuli and that ENO1 is involved in many aspects of the inflammatory response. ENO1 is known to induce inflammation through activation of the p38 MAPK and NF-κB signaling pathways [37]. Besides this, it is an effective inhibitor that downregulates the production of pro-inflammatory cytokines through the suppression of p38 MAPK and NF-κB activation following ENO1 stimulation [38]. It has also been observed that hypoxia significantly increases glucose uptake and that glucose transporter (GLUT) 1 and 3 expression are upregulated under hypoxia. Glucose metabolic enzymes GAPDH and ENO1 show upregulated expression under hypoxia [39]. The evidence indicates that the ENO1 signaling axis might serve as a potential target for the treatment of cancer. Moreover, ENO1 inhibitors can block the activation of the MAPK pathway, leading to the down-regulation of GLUT expression. Our results also confirmed the effect of CA on MAPK pathway activation and GLUT1 expression (Appendix A). Several studies have shown that ENO1 is overexpressed in many types of cancer, including breast, lung, and prostate cancers [40,41,42]. For example, estrogen promotes cell migration via the paracrine release of ENO1 from stromal cells in prostate cancer [43]. In breast cancer cells, ENO1 and other related proteins can reduce the expression of heat shock protein [44]. By contrast, ENO1 is upregulated and activated by several glucose transporters and glycolytic enzymes that are conducive to the Warburg effect [45]. Additionally, the expression of ENO1 can also affect the cell cycle [46]. ENO1 was found in the nuclei of different kinds of cell types, and it can bind to specific DNA sequences [47,48]. The silencing of the ENO1 gene by siRNA inhibits the proliferation of hepatocellular carcinoma cells, which followed a shortened S phase and an elongated G2/M phase of the cell cycle [49]. Knockdown of ENO1 expression decreases the cell growth, migration, and progression of invasion via the inactivation of the PI3K/Akt pathway in glioma cells [36]. Research has shown that ENO1 has several flexible loops in its crystal structure that can lead to different overall conformations. Furthermore, wild-type enolase cocrystallizes with magnesium (Mg), and the substrate or product adopts an entirely closed state with flexible active site loops [50]. In our study, CA participated in Michael addition with the sulfhydryl groups of ENO1 via α, β-unsaturated aldosterone. The covalent binding changed the stability and activity of ENO1 and then affected the supplementation of pyruvate. Moreover, silenced ENO1 resulted in antiproliferative effects, induced a cell cycle arrest in the G2/M/S phases, and triggered apoptosis in melanoma cells. 

Classical genetics holds that an important physiological function is usually regulated by multiple genes, and that a mutation in one of them is not fatal, but simultaneous mutation can cause death [51]. In general, the effect of small molecule medicines on gene function is concentration-dependent, and the introduction of a synergistic molecule can significantly improve the synergistic or antagonistic effect [52]. For example, the breast cancer susceptibility 2 protein (BRCA2) and poly ADP-ribose polymerase (PARP) participate in different types of DNA repair together. Using PARP inhibitors combined with BRCA2 antagonism can significantly improve the sensitivity of tumors, and this process is especially efficacious for the screening of small molecule antagonist medicines for combination therapy [53,54]. For a better antimelanoma effect, in this paper, a sublethal concentration of CA was used to find synergistic lethal medicines via a chemical genomics screening strategy. Our in vitro statistics indicated that CA combined with DTIC had the best synergistic effect of the antimelanoma medicines that are commonly used in clinic, such as FTM, TAX, TMZ and Pt. As an alkylating chemotherapeutic agent, DTIC induces DNA damage and results in growth arrest in the G2/M phase. When CA was combined with DTIC, the promoting effect drove cell cycle shortening at the G2/M phase and elongation at the S phase. The combined antineoplastic effect was reproduced in a mouse xenograft model of melanoma, and the covalent binding of CA to ENO1 was visualized in living tumor cells by an AL-CA probe. The combination of CA and DTIC effectively inhibited the replication of DNA and blocked cell division. Hence, the reduction in tumor volume was more pronounced than in any separate dosing group.

In summary, we demonstrated for the first time that a covalent inhibitor CA promotes melanoma cell apoptosis through ENO1 silencing. Furthermore, CA and DTIC have synergistic effects by changing the cell cycle arrest in vitro. These findings bring new clues to the understanding of the action of CA in antimelanoma cells. Finally, considering the drastic effect of the combination of CA with DTIC, it might be worthwhile for reducing medicine resistance and patients suffering from melanoma.

## 4. Materials and Methods

### 4.1. Reagents and Antibodies

CA (C110084) was purchased from Aladdin (Shanghai, China). 3-Phenylpropanal (BD17390) was purchased from Bidepharm (Shanghai, China). Cinnamic acid (C804991) was obtained from Macklin (Shanghai, China). FTM (S87283) and TAX (B21695) were purchased from Yuanye (Shanghai, China). DITC (F21540) was purchased from MERYER (Shanghai, China). Pt (HY-17394) was purchased from MedChemExpress (MCE, Monmouth, NJ, USA). TMZ (AK-72945) was purchased from Ark Pharm (Chicago, IL, USA). D,L-dithiothreitol was purchased from Solarbio (DTT, A285, Beijing, China). AP-III-a4 (HY-15858A) was purchased from MCE. Antibodies against Bax (50599-2-lg), Bcl_2_ (60178-1-lg), and Bid (10988-1-AP) were obtained from Proteintech (Chicago, IL, USA); ENO1 (bs-3978R), anti-GADPH (bsm-0978M), and β-actin (bs-0061R) antibodies were purchased from Bioss (Beijing, China). Alexa-Fluor-594-conjugated goat anti-rabbit IgG (ab150084) was purchased from Abcam (Cambridge, MA, UK), and a goat anti-rabbit IgG (#7074) secondary antibody was obtained from CST (Beverly, MA, USA). 

### 4.2. Cell Culture and Cell Viability Assessment

All kinds of cells were obtained from the American Type Culture Collection (Manassas, VA, USA). Among them, the A375, A549, HepG2, LA795, H9C2, and 293T cells were cultured in Dulbecco’s Modified Eagle’s Medium (DMEM, 10-013-CVR, Cellgro, Herndon, VA, USA). The A875, HeLa and SK-MEL-1 cells were cultured in Minimum Essential Medium (MEM, 10-010-CVR, Cellgro). The C918, BEAS-2B, K562, RAW264.7, HK-2, and 3T3 cells were cultured in Roswell Park Memorial Institute (RPMI)-1640 medium (C11875500BT, Gibco, Grand Island, New York, NY, USA). All cells were supplemented with 10% fetal bovine serum (FBS, 04-001-1A, Bioind, Tel Aviv, Israel), 100 units/mL penicillin (30-002-CI, Cellgro, Herndon, VA, USA), and streptomycin at 37 °C in 5% CO_2_. The Cell Counting Kit-8 (CCK8, bs-0764P, Bioss, Beijing, China) method was used to assay the cytotoxicity of CA and the other medicines. Moreover, the CI value reflecting the synergism of the two medicines was calculated using CompuSyn software.

### 4.3. Colony Formation and Cell Migration Assays

The A375 cells were plated onto six well plates (800 cells/well) and, after 24 h, CA (0–4 μM) was added to the medium and incubated for an additional 12 days. Cells were fixed in methanol for 15 min and stained with Giemsa for 30 min. The colonies were then scored and photographed. For the cell migration assay, the A375 cells (2 × 10^4^ cells/well) were placed in Transwell cell culture chambers (8 μm pore size, 3422, Corning, Shanghai, China). The cell suspension was placed in the upper chamber of the Transwell insert and incubated with CA (2.5, 5, or 10 μM), 0.5% DMSO as a negative control (vector), and TIG a as a positive control (10 μM) for 24 h. The lower chamber was then filled with chemoattractant (complete medium, 90% DMEM medium containing 10% FBS). The migrated cells were fixed with 4% formaldehyde and then stained with 0.02% crystal violet. Cells were examined, photographed, and quantified.

### 4.4. Cell Cycle and Annexin V-FITC/PI Double-Staining Assays

The A375 cells were incubated with CA (0, 20, 40, or 80 μM) for 24 h. The cells were then collected and washed in phosphate-buffered saline (PBS), after which the cells were fixed in ice-cold 70% (v/v) ethanol overnight at −20 °C. The cells were resuspended in PBS and the DNA content was used for the cell cycle assay kit (CA1510, Solarbio) to test. For the detection of apoptosis, 0, 20, 40, or 80 μM CA was added to the cell for 24 h. The cells were then collected and washed with PBS. Apoptosis was measured using the Annexin V-FITC apoptosis detection kit (A005-3, 7 Sea Pharmatech, Shanghai, China). The two experiments were detected using a flow cytometer system (BD LSR Fortessa, Indianapolis, NJ, USA). 

### 4.5. Enrichment of Target and Western Blotting Assay

In this paper, an alkynyl-CA probe (AL-CA) was synthesized and used to build Al-CA-functionalized magnetic microspheres (AL-CA-MMs) for target fishing. Cells were cultured with 10 μM Al-CA probe for 6 h. Next, 500 μL of lysis buffer (Solarbio) was added and incubated for 20 min on ice. The cell lysates were then treated with Al-CA-MMs and incubated with a catalyst system (100 μM sodium ascorbic acid and 100 μM CuSO_4_ in PBS) overnight at 4 °C. Afterward, the Al-CA-MMs were separated with magnets and washed with PBS. The enriched micro-spheres were then treated with 100 mM DTT. SDS-PAGE and Western blotting were then performed as previously described [55], and the captured protein target was recovered from the gel and sent for HPLC-mass spectrometry identification (Huada Gene Research Center, Beijing, China). 

### 4.6. Cellular Thermal Shift Assay (CETSA)

The CETSA was used to detect the stability changes of the ENO1 protein. The methods, which are described in the references, are summarized as follows: cell lysates were exposed to CA (10 μM) for 24 h at different temperatures or treated with different concentrations of CA (0.01, 0.03, 0.1, 0.3, 1,3, 10, or 30 μM) for 24 h at 69 °C, after which Western blotting was used to analyze the change in ENO1 protein [56].

### 4.7. Colocalization of Target Proteins with AL-CA Probe

The cells were treated with 10 μM CA or Al-CA probes for 6 h. Cells were then fixed with 4% paraformaldehyde and washed with PBS. After they were blocked with 10% goat serum, the cells were used for colocalization imaging. For the in vivo tumor imaging, A375 cells were subcutaneously injected into nude mice. When the tumor was approximately 0.6 mm in diameter, the mice were continuously treated with CA (60 mg/kg/day) or Al-CA (30 or 60 mg/kg/day) for two weeks. Finally, the tumor tissues were stripped, fixed, and sent for pathological sectioning. The sections were then dewaxed through a series of steps and then utilized for immunohistochemical analysis. Subsequently, an anti-ENO1 (1:1000) antibody was added to the above cells or tumor sections and incubated overnight at 4 °C. Afterward, Alexa-Fluor-594-conjugated secondary antibodies (1:1000) were added for 1 h at 37 °C. The N_3_-tag substrate (10 μM) was then added for the click reaction for 1 h at 37 °C. After washing, fluorescent images were acquired from a confocal microscope (Leica TCS SP8, Japan). For the analysis with Alexa Fluor 594, the EX WL was at 594 nm and the EM WL was at 617 nm; for the clicked fluorescence product with AL-CA, the EX WL was 488 nm and the EM WL was 520 nm.

### 4.8. RNA Interference and Quantitative Real-Time PCR (RT-PCR) Analysis

Cells were seeded into six well plates and reached 70% confluence before transfection. The siRNAs for ENO1 (siENO1, SI02654309), and scramble siRNA (siNC, SI03650318), which were purchased from Qiagen (Dusseldorf, Germany), were complexed with Lipofectamine 3000 (L3000-015, Invitrogen, Shanghai, China) based on the manufacturer’s instructions. The transfection medium was removed and replaced with a complete medium after 6 h. The expression level of ENO1 protein in the A375 cells was assessed by Western blotting after 24 h. After that, the total RNA was extracted according to the methods of the EastepTM Total RNA Super Extraction Kit (LS1040, Promega, Madison, WI, USA), and the RNA concentration was determined. The reverse transcription process was performed using the GoScriptu Reverse Transcription System (A5000, Promega). Finally, quantitative reverse transcription-polymerase chain reaction (qRT-PCR) was performed using the GoTaq quantitative polymerase chain reaction (qPCR) Master Mix (A6001, Promega, Madison, WI, USA), and ENO1 (QT00090881) and GAPDH (QT00079247) primers were designed by Qiagen (Dusseldorf, Germany). The cycling parameters for all of the genes was preheating at 95 °C for 2 min, followed by 1 cycle, followed by 40 cycles of 95 °C for 15 s, 55 °C for 1 min and 70 °C for 1 min. The data were analyzed using the 2^−ΔΔCT^ method according to the detected fluorescence intensity.

### 4.9. Xenografts In Vivo

Eight week old female BALB/c-nude mice (approximately 20 g), (specific pathogen-free, SPF) were purchased from the Laboratory Animal Experimental Center of Academy of Military Medical Sciences (SCXK2012-0004, Beijing, China). A375 cells (1 × 10^7^) were injected into nude mice. When the tumors had formed for one week, the mice were treated with DITC (25, 50, or 100 mg/kg/day) and CA (15, 30, or 60 mg/kg/day). In the combination groups, mice were treated with CA 30 mg/kg/day and DITC (25 or 50 mg/kg/day). PBS was used as a negative control. Tumor volume was measured every 3 days. Tumor volume was calculated as length × width^2^/2. After two weeks, mice were sacrificed by decollement, and tumor tissues were excised and weighed. Each experimental group consisted of six animals. Half of each tumor were fixed in 4% formaldehyde for immunohistochemistry. The other half was frozen in liquid nitrogen for subsequent use. All animal care and experimental protocols were approved by the Instructional Animal Care and Use Committee (IACUC) (DW20181223-35). 

### 4.10. Statistical Analysis

The results are expressed herein as mean values ± SD, where significant differences between two groups were analyzed by *t*-test, and the analysis of multiple groups was done by analysis of variance (ANOVA) test. Differences of *p* < 0.05 were considered statistically significant.

## 5. Conclusions

In summary, these findings demonstrated that CA can arrest the cell cycle via ENO1 and can eventually inhibit melanoma growth. Additionally, the covalent binding of CA with ENO1 changed the stability of ENO1 protein. DTIC showed a promoting effect with CA for antimelanoma both in vivo and in vitro. Additionally, this combination improved the DTIC cell cycle arrest in the S phase.

## Figures and Tables

**Figure 1 cancers-12-00311-f001:**
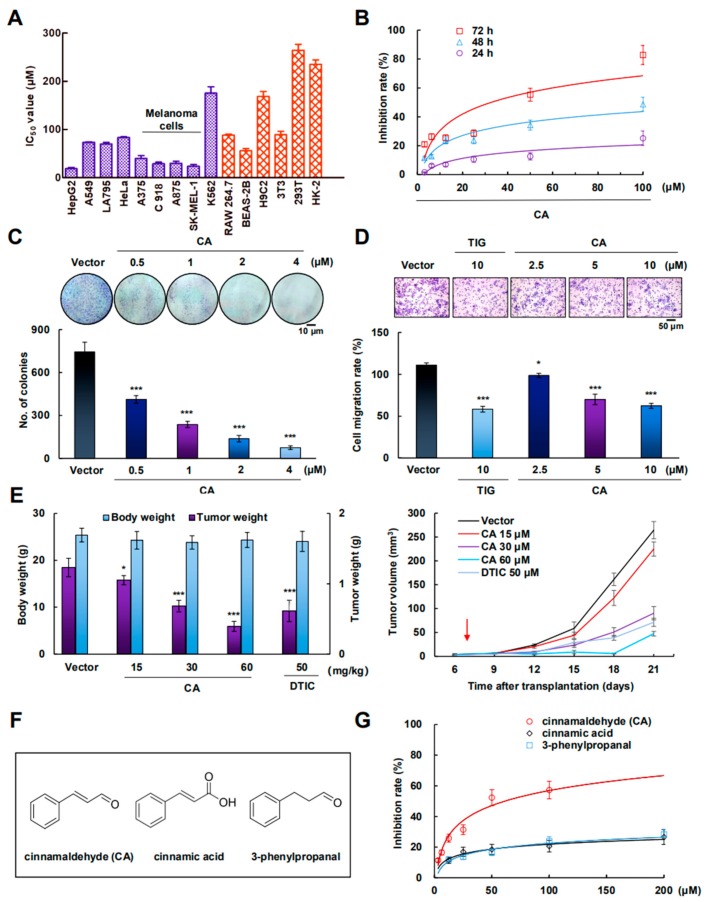
Cinnamaldehyde (CA) inhibits the proliferation of A375 cells. (**A**) The different tumor cell lines (purple) and normal cell lines (red) were treated with 0, 6.25, 12.5, 25, 50, 100, 200, or 400 μM CA for 72 h, and then IC_50_ value was assayed. (**B**) A375 cells were treated with 0, 6.25, 12.5, 25, 50, or 100 CA for 24, 48, or 72 h, and the cell viability was assayed. (**C**) A375 cells were treated with 0, 0.5, 1, 2, or 4 μM CA for 12 days, and Giemsa staining was used to conduct colony-formation assays; the histogram shows the number of colonies (*n* = 3), scale bar: 10 μm. (**D**) Cells were treated with tigecycline (TIG, 10 μM as a positive control) or CA (0, 2.5, 5, or 10 μM) for 24 h, and the migration rate was normalized by proliferation (*n* = 3), scale bar: 50 μm. (**E**) A375 cells were injected subcutaneously into nude mice. When the tumors were formed, mice were treated with dacarbazine (DTIC) (50 mg/kg/day as a positive control) and CA (15, 30, or 60 mg/kg/day) for 2 weeks (The red arrow at Day 7 represents the first day of treatment). Finally, the tumor tissues were measured, stripped, and weighed, and the tumor volume was calculated (*n* = 6). (**F**) The structures of cinnamaldehyde (CA), cinnamic acid, and 3-phenylpropanal. (**G**) A375 cells were treated with different concentrations of 3-phenylpropanal, cinnamic acid, and CA for 72 h. The cell viability was assayed. Values represent the means ± SD. * *p* < 0.05, *** *p* < 0.001 versus “vector” group.

**Figure 2 cancers-12-00311-f002:**
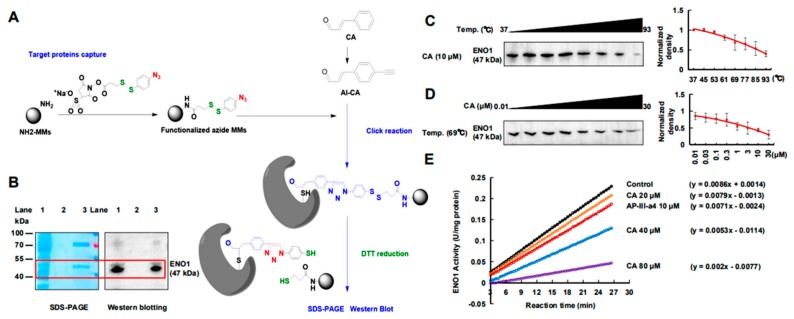
CA targeted ENO1 in A375 cells and changed its activity. (**A**) The modification process of AL-CA-MMs and the schematic diagram for target protein capture and release. (**B**) Efficiency evaluation of magnetic capture by SDS-PAGE. Lane 1 shows the A375 lysate as a loading control; Lane 2 shows the lysate captured only by the azide-modified MMs as a negative control; Lane 3 shows the lysate captured by AL-CA-MMs. Coomassie bright blue staining is shown in the left image, and the right image shows the ENO1 detected by Western blotting. (**C**) CA treatment (10 μM) decreased the thermal stability of ENO1 in cell lysates measured by the temperature-dependent Cellular thermal shift assay (CETSA) (*n* = 3). (**D**) CA treatment decreased the thermal stability of ENO1 in cell lysates measured by the concentration-dependent CETSA at 69 °C (*n* = 3). (**E**) CA inhibited the activity of ENO1. Cells were exposed to CA (0, 20, 40, or 80 μM) or a positive ENO1 inhibitor, AP-III-a4 (10 μM) for 0.5 h, and then the effect on the activity of ENO1 in lysates was determined (*n* = 3).

**Figure 3 cancers-12-00311-f003:**
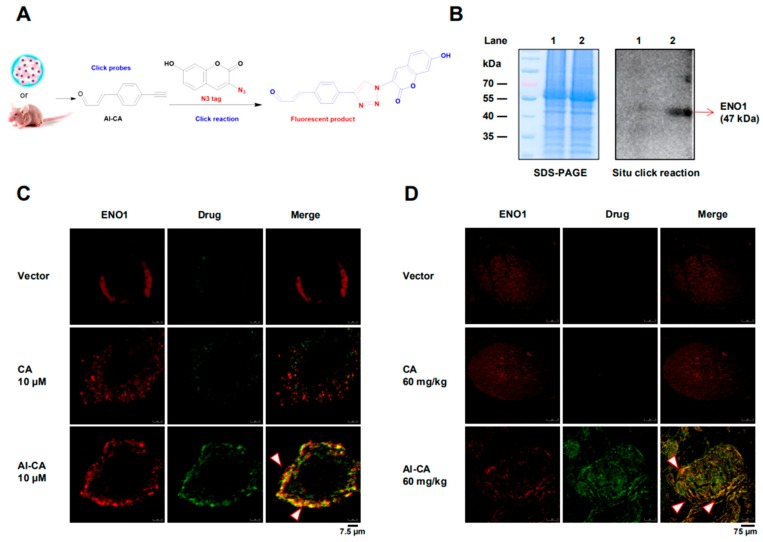
CA colocalized with ENO1 both in vitro and in vivo. (**A**) Schematic diagram of fluorescent click reaction of the Al-CA probe with the N3-tag. (**B**) In situ click reaction on PVDF membrane. After SDS-PAGE was performed, the click reaction was performed with an N3-tag in situ. Lane 1 shows the A375 lysate treated with CA as a loading control; and Lane 2 shows the A375 lysate treated with an AL-CA probe. All of the protein samples were adjusted to equal amounts before capture. (**C**) The colocalization imaging of the Al-CA probe and the ENO1 protein on A375 cells with fluorescence confocal microscopy. (**D**) In vivo imaging for CA and ENO1. A375 cells were subcutaneously injected into the nude mice. When the tumors formed for one week, mice were treated with CA (60 mg/kg/day) or Al-CA (60 mg/kg/day) for two weeks. Moreover, the tumor tissues were stripped to the pathological section for the colocalization imaging assay. The pseudo red color represents ENO1, which was stained by Alexa Fluor 594, and the Al-CA probe took on a pseudo green color via the click reaction. The yellow merger, where the arrow pointed, represents ENO1 colocalized with Al-GA.

**Figure 4 cancers-12-00311-f004:**
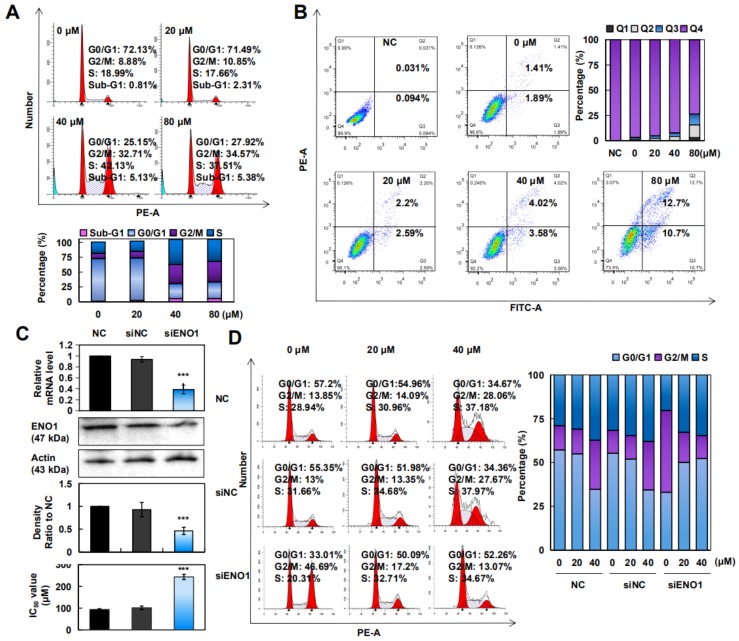
CA arrested the cell cycle, induced apoptosis, and inhibited ENO1 function. (**A**) Cells were treated with 0, 20, 40, or 80 μM CA for 24 h, after which the cells were collected and stained. The DNA content of the cells was determined with a flow cytometry system. Histograms show the percentage of cells in Sub-G1, G0/G1, G2/M, and S phase. (**B**) Cells were treated with 0, 20, 40, or 80 μM CA for 24 h. A flow cytometric analysis of CA-induced apoptosis in A375 cells was done by double-staining with Annexin V-FITC/PI. Histograms show the percentage of different regions; “Q2” and “Q3” represent late and early apoptosis, respectively. (**C**) A375 cells were transfected with siENO1 for 48 h, and the relative mRNA level of ENO1 was assayed by qRT-PCR; additionally, the expression of ENO1 in cells was assayed by Western blotting. Histograms show the intensity of the ENO1 protein bands. Finally, cell viability is presented as the IC_50_ values (*n* = 3). *** *p* < 0.001 versus “0 μM” group. (**D**) A375 cells were transfected with siENO1 for 48 h, and then cells were treated with 0, 20, or 40 μM CA for 24 h. Cells were then collected and determined via flow cytometry system. Histograms show the percentage of cells in G0/G1, G2/M, and S phase after treatment with CA. “NC” group stands for normal cultured cells without CA and interference fragment; “siNC” group was used as a negative control.

**Figure 5 cancers-12-00311-f005:**
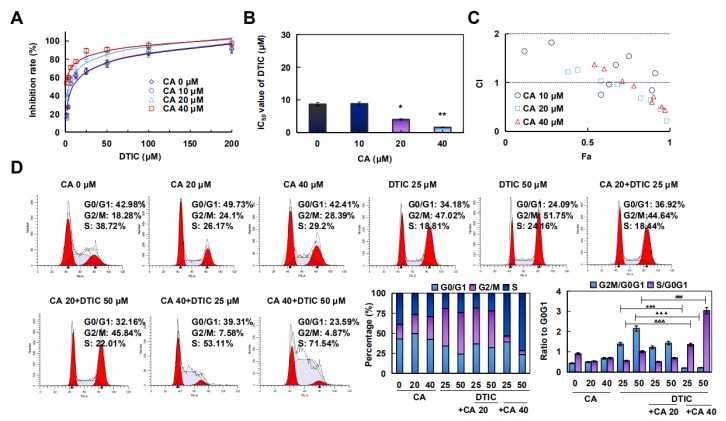
CA treatment showed a promoting effect with DTIC in vitro and in vivo. (**A**,**B**) A375 cells were treated with DTIC for 72 h, the inhibition rate was assayed, and the IC_50_ value was calculated. * *p* < 0.05, ** *p* < 0.01 versus no CA group (*n* = 3). (**C**) The cooperativity index (CI) value reflecting the synergism of two medicines was calculated using CompuSyn software. CI values of <1, 1, and >1 indicate synergistic, additive, and antagonistic effects, respectively. (**D**) Cells were treated with 0, 25, or 50 μM DTIC and with 0, 20, or 40 μM CA for 24 h and were analyzed with a flow cytometry system, and the G0/G1 ratio was calculated. *** *p* < 0.001 versus the “DTIC 25 alone” group in G2M/G0G1, ^∆∆∆^
*p* < 0.001 versus the “DTIC 25 alone” group in S/G0G1, ^▲▲▲^
*p* < 0.001 versus the “DTIC 50 alone” group in G2M/G0G1, and ^###^
*p* < 0.001 versus the “DTIC 50 alone” group in S/G0G1. (**E**) Cells were treated with TIG (10 μM as a positive control), DTIC (25 or 50 μM), or CA (20 or 40 μM) for 24 h, and the migration rate was normalized by proliferation. *** *p* < 0.001 versus the “vector” group, ^∆∆∆^
*p* < 0.001 versus the “CA 20 + DTIC 0” group, and ^##^
*p* < 0.01 versus the “CA 40 + DTIC 0” group (*n* = 3), scale bar: 50 μm. (**F**) The cells were subcutaneously injected into nude mice. When the tumors had formed for one week, mice were treated with the combination of DITC (25 or 50 mg/kg) and CA (30 mg/kg) or either medicine alone every day for two weeks (the red arrow at Day 7 was represent the first day of treatment). Average mouse weight and tumor weight are shown, and the tumor volume was calculated. * *p* < 0.05, ** *p* < 0.05, *** *p* < 0.001 versus the “vector” group, ^∆∆∆^
*p* < 0.001 versus the “DTIC 25 mg/kg” group, ^##^
*p* < 0.01 versus the “DTIC 50 mg/kg” group, and ^▲^
*p* < 0.05, ^▲▲^
*p* < 0.01 versus the “CA 30 mg/kg” group (*n* = 6).

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
