# Peer review of "Cinnamaldehyde Enhances Antimelanoma Activity through Covalently Binding ENO1 and Exhibits a Promoting Effect with Dacarbazine"

_cancers, 2020, doi:10.3390/cancers12020311_

Round 1

Reviewer 1 Report

Accept as presented

Author Response

To reviewer 1:

Comment: Accept as presented.

Response: Thank you for your recognition of our revised manuscript.

Reviewer 2 Report

I would like to thank the authors who have taken into account almost all of my previous suggestions. I write again that this study might be published, due to its high scientific quality and its potential clinical impact.

The authors have proposed responses point-by-point that I will review. Some of them are totally appropriate, but it persists few minors revisions that may be performed:

Comment 1: It seems that scientific background developed by the authors was not introduced in the revised version of the article, unfortunately. Because of specific targeted therapies used in CM, it is not a detail to have a focus on both Pi3K and MAPK pathway. I therefore suggest that this scientific backgroung developed by the authors in the responses was introduced in the article, as well as WB data. Moreover, I appreciate evaluation of p-ERK, but it lacks an evanluation of the Pi3K pathway, i.e. p-AKT and p-S6. This should be added in the final version of the article.

Comment 2: It's perfect. Thanks.

Comment 3: here there is a difference with other in vivo practices. However, I consider that presented data are highly significant and could be presented in the present form, with two minor revisions: (1) in the x axis, change "Time (day)" in "Time after transplantation (days)"; (2) indicate with an arrow at day 7 the firtst day of treatment. In my opinion, the treatment has been initiated quite too realy.

Comment 4: OK.

Comment 5: OK.

Round 2

Reviewer 2 Report

The authors have (almost) answered all requests and I think that the paper is now available for publication.

This manuscript is a resubmission of an earlier submission. The following is a list of the peer review reports and author responses from that submission.

Round 1

Reviewer 1 Report

The authors present interesting data that will be of interest to the general reader. The key issue for the authors to address is one of PK/PD in a human. What is the safe highest dose of Cinnamaldehyde that a healthy human can tolerate, and specifically, what is the plasma C max and AUC? Many natural products have a poor PK/PD profile in patients, and the authors use high micromolar concentrations of Cinnamaldehyde. Thus, without some level of understanding of "what is a clinically relevant concentration" of the agent, all of the studies presented in this manuscript may be meaningless in terms of whether the data will have relevance.

Minor point: In Figure 4, the cell cycle analyses do not appear to show any sub-G1 fragments yet the flow / viability data show cell death.

Reviewer 2 Report

The study presented by Weiyi Zhang is of high interest, suggesting a novel therapeutic approach of skin melanoma (SM). Various experiments have been performed to demonstrate a therapeutic potential of Cinnamaldehyde (CA), both in vitro and in vivo, with a focus to define mechanism of the antitumor activity. Hence, this study might be favorably considered for a further publication.

However, various major issues should be mentioned, in regards to different lacks and, mainly, to inappropriate presentation of the data. Those issues could be summarized, as follow:

In the introduction, almost no notification was made of BRAF mutations observed in a half of SM patients. This absence is consequeltly observed in all the paper where no study was performed with BRAF inhibitors. This point is a clear inadequacy, as no study of the impact of CA on both MAPK nor Pi3K pathways have been realized. Due to the mechanism of ENO1, it would be of high interest to evaluate Pi3K function under therapies, such as GLUT1 expression, and so on, as well as of MAPK (and this because of the high interactions between these two signaling pathways). In the paragraph 2.1, it is not possible to mention a "specificity" of CA against melanoma cells. Indeed, all tested cell lines are melanoma cells. In the same paragraph, it was mentioned that CA was most efficient on the A375 cells. However, in the figure 1A, the IC50 was about 70 µM, whereas it was about 20 µM for the three other cell lines. In both Figures 1 and 5, in vivo experiments might be presented with curves of tumor growth. I may suggest to evaluate probability of tumor progression, as reported in the now-referenced publication of Gao et al (Nature Med 2015). Finally, the authors mentioned a synergistic activity between CA and DTIC. They should, for in vitro data, present Loewe or similar synergistic figures; moreover, for in vivo data, clear statistical analysis should be performed to compare DTIC alone, CA alone and combination. The term "synergy" should be used cautiously with in vivo experiments. It lacks statistical data for in vitro study evaluating cell cycle. I may suggest to combine G2/M and S versus G0/G1 phases.

Overall, this study is of high interest but might be strongly improved. The revised version of the paper might then be reevaluated.